# Effect of senior executives' overseas experience on corporate green innovation

**Fangze Cheng**[1], **Xin Kuang**[2]*

1 Business School, Chengdu University, Chengdu, Sichuan, China, 2 Institute of Chinese Financial Studies, Southwestern University of Finance and Economics, Chengdu, Sichuan, China

* xkuang@swufe.edu.cn

**Data Availability Statement:** All relevant data are within the manuscript and its Supporting information files.

**Funding:** The author(s) received no specific funding for this work.

## Abstract

Against the background of increasingly severe environmental problems, green development has gained widespread attention, and green innovation has thus become crucial for enterprises. This study used 2007–2019 data from listed A-share companies in China to evaluate the effect of senior executives' overseas experience on corporate green innovation. The results showed that senior executives' overseas experience could promote green innovation in companies. This positive effect was more significant for private enterprises and high-tech enterprises, especially in eastern China. The CEO pay regulation have a significant negative moderating effect on this positive effect. This study enriches upper echelons theory and provides theoretical support for government agencies to accelerate innovative green development strategies. The results can also provide a decision-making basis for governments to formulate policies to promote enterprises' green development.

## Introduction

The past 40 years have seen increases in extreme climate disasters around the world. Many countries have recognized the need to switch to green economic development, based on the goals of saving energy and reducing emissions. The key to such transformation lies in using effective measures to promote enterprises' green innovation. For emerging economies, knowledge transfer and spillover is a significant factor promoting innovation and market economy development [1]. Executives with experience working or studying overseas can promote green innovation through the intermediary of knowledge transfer and spillover [2]. Many studies have investigated the effect of executives' personal characteristics and backgrounds on corporate innovation [3–6]. Few, however, have specifically examined the relationship between executives' overseas experience and corporate green innovation. It should also be noted that currently the academic community mostly studies the two (green and innovation) separately, without analyzing them in a unified structure.

We innovatively combine green and innovation to study the impact of senior executives' characteristics on corporate green innovation behavior. Specifically, we selected A-share companies listed on China's Shanghai and Shenzhen exchanges from 2007 to 2019 to evaluate the effect of senior executives' overseas experience on corporate green innovation and explore the effects under different scenarios. We found that senior executives' overseas experience could

**Competing interests:** The authors have declared that no competing interests exist.

promote green innovation in companies. This positive effect was more significant for private enterprises and high-tech enterprises, especially in eastern China. Furthermore, we found that CEO pay regulation have a negative moderating effect on the promotion of green innovation by senior executives' overseas experience.

Our work has at least a few contributions. Our perspective is focused on green innovation, which is currently receiving widespread attention from the academic community, and we conduct rich heterogeneity analysis. And we explored the moderating effect of CEO pay regulation. This provides a reference for studying enterprise innovation behavior. On the other hand, this suggests that we need to focus on the impact of company executives on green innovation. Of course, this study similarly helps to enrich upper echelons theory and provides theoretical support for governments to accelerate the implementation of green development strategies. Furthermore, this research can provide a decision-making basis for governments to formulate policies to promote enterprises' green development.

Our next structure mainly includes literature review and hypothesis development, method, results and analysis, examination of subsamples, moderating effect analysis, robustness test, as well as conclusions and policy recommendations.

## Literature review and hypothesis development

### Executives' overseas experience and enterprises' green innovation

In China, having overseas experience tends to be viewed as a sign of being multicultural, having a good educational background, and possessing robust vocational skills. Executives with experience abroad have unique social capital (e.g., overseas relationships, information networks) and human capital (e.g., advanced knowledge and an international perspective). Upper echelons theory holds that the personal characteristics and heterogeneity of a company's management (e.g., age, educational background, career path) can influence a firm's business decisions and financial behaviors [7]. In addition, the imprinting theory explains that the personal experiences of executives are the key to influencing corporate decision-making [8–10]. Both upper echelons theory and imprinting theory can be used to study the behavior of companies, but the former is mainly used to explain the influence of executives. Senior executives' overseas experience can promote a company's green innovation in the following ways: 1) executives' overseas experience can produce a collision of different cultures and knowledge. A diverse executive team can lend a company richer cognitive resources and provide strong motivation for organizational innovation, change, performance, and strategic decision-making [11, 12]. Alexiev et al. [13] confirmed that when the senior management has diverse work experience and educational backgrounds, it can significantly increase innovation inputs. Optimizing the structure of the senior management team can also help increase the innovation input of the enterprise. 2) Senior executives' overseas experience can have knowledge and experience spillover effects, serving to communicate cutting-edge green innovation technologies, alleviate firms' knowledge and technology resource constraints, and thus enhance green innovation capabilities. 3) In developed regions such as the United States and Europe, laws and regulations tend to be more scientific, government supervision is more effective, and the awareness of legal and social responsibility permeates all aspects of cultural life. Therefore, executives with overseas experience in such areas will be influenced by their laws and sense of social responsibility, causing them to pay more attention to corporate social responsibility, which will in turn promote green innovation.

Based on the above, the following is proposed:

**Hypothesis 1.** With other influencing factors controlled (or stable), senior executives' overseas experience positively correlates with a company's green innovation.

## Property rights

Enterprise ownership largely determines the distribution of resources and the corporate governance structure [14–16]. When a company is government owned, it generally lacks incentives to innovate [17]. Therefore, considering the multiple ownership structures found in China, it is necessary to distinguish the different effects of public and nonpublic ownership. For Chinese state-owned enterprises (SOEs), the goal is not to maximize corporate interest but to assume more public functions. Thus, SOEs tend to have less incentive and demand for innovation. Furthermore, the managers of SOEs tend to have political incentives for promotion as opposed to economic incentives to achieve business results, causing these managers to adopt the viewpoint of "not seeking merit but seeking no fault, which will also negatively affect innovation.

The following hypothesis is therefore proposed:

**Hypothesis 2.** With other factors controlled, the effect of managers' overseas experience on green innovation will be more significant in Non-SOEs than in SOEs.

## Regional differences

Regarding the differences in resource endowments and cultural traditions in China's various provinces, there are large imbalances in economic development among different provinces, which might have an interactive effect on the relationship between executives' overseas experience and corporate green innovation. First, the regional culture will directly affect the innovation ability of executives with overseas experience. Second, there are major differences in the policy support available to overseas returnees in different regions. Compared with the inland provinces, the coastal areas opened up to the world much earlier, have higher levels of development, possess a better entrepreneurial environment, and enjoy preferential policies [18]. China therefore presents an unbalanced talent innovation environment that is stronger in the east and weaker in the west. With the strong, policy-supported entrepreneurial environment in the coastal areas, returnees can better utilize their social capital advantages.

Therefore, the following is proposed:

**Hypothesis 3.** With other factors controlled, executives' overseas experience has a more significant effect on enterprises' green innovation in the eastern region than in the central and western regions.

## High-tech industry

The effect of executives' overseas experience on corporate green innovation may vary greatly in different industries. For high-tech industries, which are highly competitive and rely heavily on technological iteration to gain market position, the spillover effects of knowledge and skills brought about by managers' overseas experience will be more fully utilized. For non-high-tech companies, their requirements for the spillover effects of overseas knowledge and skills are not as strong. Their managers can take relatively less advantage of ingenuity gained from overseas experience, and the promotion of innovation is relatively weak.

The following is therefore proposed:

**Hypothesis 4.** With other factors controlled, executives' overseas experience has a more significant effect on green innovation in high-tech enterprises than in non-high-tech enterprises.

## Method

### Samples and data

Listed Chinese companies began disclosing patent data in 1990, but it was not until 2007 that green patent disclosure data were improved. Therefore, this study took A-share companies listed on the Shanghai and Shenzhen exchanges that published patent information from 2007 to 2019 as the initial sample. Samples were then screened as follows: 1) companies in the financial industry, 2) companies with missing data or noncontinuous disclosure, and 3) companies with abnormalities in their ST, *ST, and financial indicators were excluded. Finally, 15334 panel data of 2681 sample companies were obtained. It should be noted that some variables have missing observations, so not all variables have observations of 15334, as shown in Table 2. Overseas experience data for executives came from the personal characteristics of executives given in the CSMAR database. Missing information was manually compiled and supplemented based on executives' resumes disclosed in annual reports. Data on enterprises' green patent applications and citations were obtained from the CSMAR and Wind databases and were supplemented by manual collation from companies' annual reports. The remaining data were selected from the CSMAR and Wind databases. To avoid the influence of extreme values, the main continuous variables were winsorized at 1% from beginning to end.

### Major variables construction

**Dependent variables.** Multiple factors were considered, including the number of green patent applications (*Patent*) and green patent references (*Citation*). There are currently two main methods that researchers use to measure the innovation activities of enterprises: one is the input method, which measures the amount of enterprise R&D investment; the other is the output method, which measures the patent applications and citations of an enterprise in the current year. As the most important measure of a company's innovation output, patents can more fully reflect the utilization and transformation of enterprise R&D investment. Therefore, the number of applications and the number of citations of corporate green patents were selected as the dependent variables.

**Independent variables.** Three methods are mainly used to measure executives' overseas experience: 1) construct dummy variables. When at least one of the executives has experience studying or working overseas, the value is 1; otherwise, 0. 2) take the sum of the number of corporate executives who have overseas experience. 3) divide the number of returnee executives in a company by the total number of executives. In the main regression analysis, we used the dummy variable *Oversea_Board*. In addition, we conducted robustness tests using the number variable *OverseaNum* and the ratio variable *OverseaRatio*. The term executives generally refers to people working in important positions in a listed company who have a significant influence on corporate decision-making and performance. At present, there is no uniform standard in the research for recognizing senior management and senior management teams. In this study, senior management refers to the board of directors, board of supervisors, and senior management (such as the Chief Executive Officer, Chief Financial Officer, Chief Operating Officer) disclosed in listed companies' annual reports. This standard complies with China's "Company Law" regulation.

## Model development

To test the effect of senior executives' overseas experience on corporate green innovation, the following model was developed:

$$Patent_{i,t}, Citation_{i,t} = \beta_0 + \beta_1 Oversea\_Board_{i,t} + \theta X + \omega_{i,t} + \eta_t + \epsilon_{i,t}, \tag{1}$$

In Eq (1), $i$ and $t$ represent company and year respectively, and $\beta_0$ is a constant term. The dependent variable is $Patent_{i,t}(Citation_{i,t})$, the independent variable is $Oversea\_Board_{i,t}$, and $X$ is a series of control variables. $\omega_{i,t}$ is the industry fixed effect of the companies, $\eta_t$ is the year fixed effect, and $\epsilon_{i,t}$ is the error term.

Table 1 shows the definitions of the variables in Eq (1). Among them, *Patent* and *Citation* refer to the corporate green innovation indicators, which measure the "quantity" and "quality" of a company's green innovation, respectively. *Oversea_Board* is a measure of executives' overseas experience. *X* is a series of variables, selected following Pittman and Fortin [19].

The model mainly focuses on the regression coefficient of executives' overseas experience (*Oversea_Board*). If the regression coefficient is significantly positive, it means executives' overseas experience promotes the company's green innovation, thus supporting **Hypothesis 1**. Considering that a possible correlation exists between executives' overseas experience and corporate green innovation at the company level, to avoid the company-level clustering effect on the standard error, company-level clustering adjustment was applied to the standard deviation. Based on the main regression model, the samples were grouped and regressed according to enterprise ownership, enterprise regional distribution, and technological attributes to test **Hypothesis 2**, **Hypothesis 3**, and **Hypothesis 4**.

## Results and analysis

### Descriptive statistics

Table 2 shows the descriptive statistical results. The medians of *Patent* and *Citation* were 1.099 and 2.197, respectively, and the averages of the two were 1.310 and 2.485, respectively. We can see that there is a certain degree of bias in the patent data. The rest are control variables, and the selection criteria followed Pittman and Fortin [19]. It was found that companies with green innovation were relatively large in scale, low in debt ratios, high in operating income

**Table 1. Variable definitions.**

| Variable type | Variable symbol | Variable definitions |
|---|---|---|
| Dependent variables | *Patent* | Ln (number of company's green patent applications + 1). |
| | *Citation* | Ln (number of corporate green patent citations + 1). |
| Independent variable | $Oversea_Board$ | 1 if the company has at least one senior executive with overseas experience; 0 otherwise. |
| Control variables | *Size* | Ln (average total assets). |
| | *Lev* | Total liabilities divided by total assets. |
| | *ROA* | Net profit divided by total assets. |
| | *Liquid* | Current assets divided by total assets. |
| | *Revg* | Operating income growth rate. |
| | *TBQ* | The ratio of market value to asset replacement cost. |
| | *Inshare* | Institutional investor shareholding ratio. |
| | *Indep* | Independent directors divided by number of directors. |
| | *DirAge* | Average age of executives. |

**Table 2. Descriptive statistics of the main variables.** This table reports the summary statistics. Variables are defined in Table 1. The sample period is from 2007 to 2019.

| Variable | mean | sd | min | p50 | max | N |
|---|---|---|---|---|---|---|
| Patent | 1.310 | 1.322 | 0.000 | 1.099 | 5.273 | 15334 |
| Citation | 2.485 | 1.465 | 0.000 | 2.197 | 6.951 | 11314 |
| Oversea_Board | 0.115 | 0.319 | 0.000 | 0.000 | 1.000 | 15334 |
| Size | 22.250 | 0.568 | 21.600 | 22.210 | 22.930 | 15334 |
| Lev | 0.456 | 0.204 | 0.066 | 0.453 | 0.943 | 15334 |
| ROA | 0.034 | 0.069 | −0.343 | 0.035 | 0.192 | 15334 |
| Liquid | 0.565 | 0.201 | 0.097 | 0.580 | 0.944 | 15334 |
| Revg | 0.185 | 0.417 | −0.544 | 0.117 | 2.667 | 14772 |
| TBQ | 2.137 | 1.299 | 0.900 | 1.728 | 8.436 | 14971 |
| Inshare | 0.066 | 0.071 | 0.000 | 0.042 | 0.331 | 15334 |
| Indep | 0.374 | 0.054 | 0.333 | 0.333 | 0.571 | 15305 |
| DirAge | 51.640 | 8.368 | 33.000 | 51.000 | 74.000 | 15332 |

growth rates, and high in interest rate growth rates. This indicates that listed Chinese companies with green innovation were relatively high-quality companies.

Table 3 reports the correlation matrix. The correlation coefficient between *Patent*(*Citation*) and *Oversea_Board* was 0.028(0.036) and was highly significant. It shows that the overseas experience of executives can significantly enhance the company's level of green innovation. This preliminarily confirmed our **Hypothesis 1**.

## Basic regression results

Table 4 shows the results of the effect of executives' overseas experience on corporate green innovation. The regression results in columns (1) and (4) do not involve any control variables, those in columns (2) and (5) control industry and year effects, and those in columns (3) and (6) further control the control variables that measure enterprise characteristics. The regression results in column (3) show that the *Oversea_Board* coefficient is significantly positive at the 5% level. This indicates that compared with companies with no overseas experience, the number of green patents for those with overseas experience was significantly higher. The regression result in (6) shows that the *Oversea_Board* coefficient is significantly positive at the 1% level. This indicates that compared with companies with no overseas experience, the number of green patent citations for those with overseas experience was significantly higher. The regression results show that, regardless of the quantity or quality of green patents, executives' overseas experience had a significant positive effect on corporate green innovation. The overseas experience of executives endows them with more advanced business concepts, environmental awareness, and innovative spirit. The characteristics of executives have a significant impact on the company's behavior and performance according to the upper echelons theory, thus promoting green innovation in the company. Thus, **Hypothesis 1** is supported.

In addition, the estimated coefficient of enterprise size was significantly positive, indicating that the larger the enterprise, the higher the quantity and quality of its green innovation. The estimated coefficient of corporate financial liquidity (*Liquid*) was also significantly positive, indicating that the better the corporate liquidity, the stronger the company's green innovation motivation. These results support the study hypotheses and are also consistent with the results of previous research. It was also found that the coefficient of *TBQ* (Tobin's Q value) was negative and significant. Thus, the greater the company's book-to-market value ratio, the lower the efficiency and quality of green innovation.

**Table 3. Correlation matrix.** This table reports the correlation matrix. Variables are defined in Table 1. The sample period is from 2007 to 2019.

| | 1 | 2 | 3 | 4 | 5 | 6 |
|---|---|---|---|---|---|---|
| 1 Patent | 1 | | | | | |
| 2 Cition | 0.566*** | 1 | | | | |
| 3 Oversea_Board | 0.028*** | 0.036*** | 1 | | | |
| 4 Size | 0.290*** | 0.282*** | 0.019** | 1 | | |
| 5 Lev | 0.115*** | 0.127*** | 0.014* | 0.452*** | 1 | |
| 6 ROA | 0.058*** | -0.026*** | 0.010 | 0.015* | -0.366*** | 1 |
| 7 Liquid | 0.035*** | 0.011 | 0.000 | -0.238*** | -0.114*** | 0.127*** |
| 8 Revg | 0.039*** | -0.038*** | 0.002 | 0.037*** | 0.001 | 0.233*** |
| 9 TBQ | -0.130*** | -0.088*** | 0.014* | -0.370*** | -0.270*** | 0.087*** |
| 10 Inshare | 0.121*** | 0.069*** | 0.038*** | 0.192*** | 0.004 | 0.202*** |
| 11 Indep | 0.051*** | 0.051*** | 0.016* | 0.020** | 0.021*** | -0.039*** |
| 12 DirAge | 0.062*** | 0.081*** | 0.003 | 0.116*** | 0.032*** | 0.007 |
| | 7 | 8 | 9 | 10 | 11 | 12 |
| 1 Patent | | | | | | |
| 2 Cition | | | | | | |
| 3 Oversea_Board | | | | | | |
| 4 Size | | | | | | |
| 5 Lev | | | | | | |
| 6 ROA | | | | | | |
| 7 Liquid | 1 | | | | | |
| 8 Revg | 0.047*** | 1 | | | | |
| 9 TBQ | 0.071*** | 0.034*** | 1 | | | |
| 10 Inshare | 0.030*** | 0.100*** | 0.178*** | 1 | | |
| 11 Indep | 0.035*** | -0.005 | 0.012 | -0.016** | 1 | |
| 12 DirAge | -0.029*** | -0.022*** | -0.048*** | -0.005 | 0.049*** | 1 |

To further explore the relationship between executives' overseas experience and corporate innovation, a heterogeneous study was further conducted based on the characteristics of executives with overseas experience. Table 5 shows the regression results grouped according to the senior management position category. If the executive was the chairman or general manager, that person was regarded as a senior manager (*Oversea_Board_S*); other senior managers were regarded as junior managers (*Oversa_Boaed_J*). Columns (1) and (2) are the regression results for a sample of companies with junior managers as the executives with overseas experience. The *Oversea_Board_J* coefficient was positive but not significant. This indicates that for the sample of companies with junior managers with overseas experience, overseas experience did not significantly promote enterprises' green innovation. Columns (3) and (4) are the regression results for the sample of companies with senior managers who had overseas experience. The *Oversea_Board_S* coefficient was positive at 10% and 1%. This indicates that for the sample of companies with senior managers with overseas experience, overseas experience had a significant effect on promoting enterprises' green innovation. Since the chairman or CEO often has greater decision-making power, their overseas experience may have different effects on a company's investment efficiency compared to other executives.

Executives' overseas experience can be divided into overseas education experience and overseas work experience. Different types of experiences may have different effects on the promotion of corporate green innovation. Table 6 shows the regression of sample groups according to the type of experience. Columns (1) and (2) are the regression results for the samples

**Table 4. Main regression results for the effect of returnees' experience on corporate green innovation.**

| | (1) | (2) | (3) | (4) | (5) | (6) |
|---|---|---|---|---|---|---|
| | *Patent* | *Patent* | *Patent* | *Citation* | *Citation* | *Citation* |
| Oversea_Board | 0.115*** | 0.164*** | 0.094** | 0.169*** | 0.254*** | 0.167*** |
| | (3.26) | (3.54) | (2.20) | (3.57) | (4.12) | (2.91) |
| Size | | | 0.634*** | | | 0.659*** |
| | | | (14.04) | | | (10.08) |
| Lev | | | 0.206* | | | 0.393** |
| | | | (1.76) | | | (2.39) |
| ROA | | | 1.025*** | | | 0.152 |
| | | | (4.62) | | | (0.50) |
| Liquid | | | 0.491*** | | | 0.421** |
| | | | (3.88) | | | (2.43) |
| Revg | | | 0.0182 | | | -0.200*** |
| | | | (0.61) | | | (-5.76) |
| TBQ | | | -0.064*** | | | -0.003 |
| | | | (-4.75) | | | (-0.14) |
| Inshare | | | 1.454*** | | | 0.892** |
| | | | (5.14) | | | (2.45) |
| Indep | | | 1.076*** | | | 0.864* |
| | | | (2.88) | | | (1.76) |
| DirAge | | | 0.004** | | | 0.007*** |
| | | | (2.41) | | | (3.30) |
| Constant | Yes | Yes | Yes | Yes | Yes | Yes |
| Year fixed effects | No | Yes | Yes | No | Yes | Yes |
| Industry fixed effects | No | Yes | Yes | No | Yes | Yes |
| r2_a | 0.00 | 0.06 | 0.16 | 0.00 | 0.09 | 0.17 |
| F | 10.66 | 12.55 | 40.86 | 12.71 | 16.99 | 19.89 |
| N | 15334 | 15334 | 14413 | 11314 | 11314 | 10912 |

This table reports the results of OLS regressions analyzing the effect of returnees' experience on corporate green innovation. Variables are defined in Table 1. The sample period is from 2007 to 2019. In parentheses are t-statistics.

***, **, * indicate statistical significance at the 1%, 5%, and 10% level, respectively.

based on executives' overseas education experience. The *Oversea_Board* coefficient was positive. However, it was only significant for the number of patents at the 1% level, while the number of patent citations was not significant. This indicates that the effect of overseas education experience on corporate green innovation was only reflected in the "quantity" aspect. Columns (3) and (4) are the regression results for the samples based on overseas work experience. The *Oversea_Board* coefficients were positive at 1% and 5%, indicating that the "quantity" and "quality" of executives' overseas work experience significantly promoted corporate green innovation. The regression results in Table 6 show that senior executives' overseas work experience improved corporate green innovation more so than overseas education experience.

## Examination of subsamples

To verify **Hypothesis 2–4**, the samples were grouped according to enterprise ownership, regional distribution, and technological attributes, and regression was performed based on Eq (1).

**Table 5. Effect on green innovation according to executive position.**

| | (1) | (2) | (3) | (4) |
|---|---|---|---|---|
| | Junior manager | | Senior manager | |
| | *Patent* | *Citation* | *Patent* | *Citation* |
| *Oversa_Boaed_J* | 0.141 | 0.176 | | |
| | (1.63) | (1.24) | | |
| *Oversa_Boaed_S* | | | 0.081* | 0.158*** |
| | | | (1.87) | (2.74) |
| Control variables | Yes | Yes | Yes | Yes |
| *Constant* | Yes | Yes | Yes | Yes |
| Year fixed effects | Yes | Yes | Yes | Yes |
| Industry fixed effects | Yes | Yes | Yes | Yes |
| r2_a | 0.16 | 0.17 | 0.16 | 0.17 |
| F | 41.34 | 20.07 | 40.96 | 19.77 |
| *N* | 14413 | 10912 | 14413 | 10912 |

This table reports the results of OLS regressions analyzing the effect of returnees' experience on corporate green innovation according to executive position. Variables are defined in Table 1. The sample period is from 2007 to 2019. In parentheses are t-statistics.

***, **, * indicate statistical significance at the 1%, 5%, and 10% level, respectively.

Table 7 shows the regression results grouped by enterprise ownership. Columns (1) and (3) are the regression results for the sample of state-owned enterprises (SOE), and the coefficients of *Oversea_Board* are both positive and insignificant. Columns (2) and (4) are the regression results for the private enterprise samples (Non-SOE), and the coefficients of *Oversea_Board* are significantly positive at the levels of 10% and 5%, respectively. The results in Table 7 show that the effect of senior executives' overseas experience on promoting corporate green innovation mainly existed for private companies. Kim and Mahoney [20] found that the effect of

**Table 6. Effect on corporate green innovation based on type of overseas experience.**

| | (1) | (2) | (3) | (4) |
|---|---|---|---|---|
| | Educational experience | | Work experience | |
| | *Patent* | *Citation* | *Patent* | *Citation* |
| *Oversea_Edu* | 0.241*** | 0.050 | | |
| | (2.60) | (0.37) | | |
| *Oversea_Work* | | | 0.082* | 0.129** |
| | | | (1.85) | (2.23) |
| Control variables | Yes | Yes | Yes | Yes |
| *Constant* | Yes | Yes | Yes | Yes |
| Year fixed effects | Yes | Yes | Yes | Yes |
| Industry fixed effects | Yes | Yes | Yes | Yes |
| r2_a | 0.16 | 0.17 | 0.16 | 0.17 |
| F | 42.21 | 19.80 | 40.83 | 19.76 |
| *N* | 14413 | 10912 | 14413 | 10912 |

This table reports the results of OLS regressions analyzing the effect of returnees' experience on corporate green innovation based on type of overseas experience. Variables are defined in Table 1. The sample period is from 2007 to 2019. In parentheses are t-statistics.

***, **, * indicate statistical significance at the 1%, 5%, and 10% level, respectively.

**Table 7. Heterogeneity analysis according to enterprise ownership.**

|  | (1) | (2) | (3) | (4) |
|---|---|---|---|---|
|  | SOE | Non-SOE | SOE | Non-SOE |
|  | *Patent* | *Patent* | *Citation* | *Citation* |
| *Oversea_Board* | 0.121 | 0.085* | 0.247 | 0.163** |
|  | (1.60) | (1.70) | (1.65) | (2.35) |
| Control variables | Yes | Yes | Yes | Yes |
| *Constant* | Yes | Yes | Yes | Yes |
| Year fixed effects | Yes | Yes | Yes | Yes |
| Industry fixed effects | Yes | Yes | Yes | Yes |
| r2_a | 0.20 | 0.14 | 0.21 | 0.16 |
| F | 17.71 | 23.55 | 6.61 | 13.95 |
| N | 6045 | 8376 | 4834 | 6078 |

This table reports the results of OLS regressions analyzing the effect of returnees' experience on corporate green innovation according to heterogeneity of enterprise ownership. Variables are defined in Table 1. The sample period is from 2007 to 2019. In parentheses are t-statistics.

***, **, * indicate statistical significance at the 1%, 5%, and 10% level, respectively.

managerial background characteristics on enterprise overinvestment differed among enterprises with different ownership types. Ahmed and Duellman [21] also found that the effects of managers' educational background, age, tenure, and other characteristics on corporate accounting conservatism differed among companies with different property rights. O'Connor et al. [22] suggested that the managers of SOEs face political promotion pressures, thus leading to more cautious management behavior. In Non-SOEs, the high concentration of equity allows corporate executives—especially the chairman—greater decision-making space. The underdeveloped professional manager market and the high level of information asymmetry have led to a large amount of nepotism in Non-SOEs and combined chairman and general manager roles. The chairman of a private enterprise generally has the dual identity of an owner and a decision-maker, often personally handling the important matters of the enterprise. Furthermore, under the imperfect corporate governance mechanism of China's enterprises, the personal characteristics of senior executives have a significant effect on business decision-making in Non-SOEs.

Table 8 shows the regression results grouped by enterprise region. Following the regional division method determined by the National Bureau of Statistics of China in 2011, the sample was divided into the eastern region and the central and western regions. Columns (1) and (3) are the regression results for the samples in the eastern region, and the coefficients of *Oversea_Board* are significantly positive at the levels of 1% and 5%, respectively. Columns (2) and (4) are the regression results for enterprises in the central and western regions. The coefficient of *Oversea_Board* is positive but not significant. Table 8 shows that the effect of executives' overseas experience on promoting green innovation mainly existed in companies in the eastern region. Since the eastern region has been economically open for longer and has a higher degree of marketization, it is considered more developed than the central and western regions in terms of the legal environment, administrative efficiency, and financial market effectiveness. The good institutional conditions in the east are beneficial for returnee executives and allow human capital to play a greater role.

Table 9 shows the regression results grouped by enterprise technology attributes. Following the industry classifications in the 2012 China Securities Regulatory Commission's Industry Classification Guidelines for Listed Companies, the samples were divided into high-tech and

**Table 8. Heterogeneity analysis according to enterprise region.**

|  | (1) | (2) | (3) | (4) |
|---|---|---|---|---|
|  | **Eastern** | **Central–western** | **Eastern** | **Central–western** |
|  | *Patent* | *Patent* | *Citation* | *Citation* |
| *Oversea_Board* | 0.086* | 0.012 | 0.141** | 0.123 |
|  | (1.71) | (0.17) | (2.13) | (1.16) |
| Control variables | Yes | Yes | Yes | Yes |
| *Constant* | Yes | Yes | Yes | Yes |
| Year fixed effects | Yes | Yes | Yes | Yes |
| Industry fixed effects | Yes | Yes | Yes | Yes |
| r2_a | 0.18 | 0.15 | 0.18 | 0.17 |
| F | 30.53 | 12.47 | 16.34 | 5.62 |
| N | 10055 | 4358 | 7642 | 3270 |

This table reports the results of OLS regressions analyzing the effect of returnees' experience on corporate green innovation according to heterogeneity of enterprise region. Variables are defined in Table 1. The sample period is from 2007 to 2019. In parentheses are t-statistics.

***, **, * indicate statistical significance at the 1%, 5%, and 10% level, respectively.

non-high-tech industries. Columns (1) and (3) are the regression results for the sample of high-tech companies; the coefficients of *Oversea_Board* are significantly positive at the 5% and 1% levels, respectively. Columns (2) and (4) are the regression results for non-high-tech companies. The coefficient of *Oversea_Board* is positive but not significant. The results in Table 9 indicate that the effect of executives' overseas experience on promoting corporate green innovation mainly existed in high-tech companies. High-tech companies have better corporate governance mechanisms than non-high-tech companies. Therefore, executives' overseas experience is more likely to have spillover effects in high-tech firms.

This series of heterogeneity analyses confirms **Hypothesis 2–4**.

**Table 9. Heterogeneity analysis based on high-tech and non-high-tech enterprises.**

|  | (1) | (2) | (3) | (4) |
|---|---|---|---|---|
|  | **High-tech** | **Non-high-tech** | **High-tech** | **Non-high-tech** |
|  | *Patent* | *Patent* | *Citation* | *Citation* |
| *Oversea_Board* | 0.140** | 0.042 | 0.330*** | 0.054 |
|  | (2.09) | (0.78) | (3.73) | (0.75) |
| Control variables | Yes | Yes | Yes | Yes |
| *Constant* | Yes | Yes | Yes | Yes |
| Year fixed effects | Yes | Yes | Yes | Yes |
| Industry fixed effects | Yes | Yes | Yes | Yes |
| r2_a | 0.12 | 0.19 | 0.13 | 0.20 |
| F | 22.34 | 32.51 | 6.28 | 22.14 |
| N | 6367 | 8046 | 4662 | 6250 |

This table reports the results of OLS regressions analyzing the effect of returnees' experience on corporate green innovation based on high-tech and non-high-tech enterprises. Variables are defined in Table 1. The sample period is from 2007 to 2019. In parentheses are t-statistics.

***, **, * indicate statistical significance at the 1%, 5%, and 10% level, respectively.

**Table 10. The moderating role of CEO pay regulation.**

|  | (1) | (2) |
| --- | --- | --- |
|  | *Patent* | *Citation* |
| *Oversea_Board* | 1.140*** | 0.875*** |
|  | (5.34) | (3.00) |
| *Rest* | -0.694*** | -0.808** |
|  | (-6.20) | (-2.20) |
| *Oversea_Board × Rest* | -0.767* | -1.413** |
|  | (-1.72) | (-2.27) |
| Control variables | Yes | Yes |
| *Constant* | Yes | Yes |
| Year fixed effects | Yes | Yes |
| r2_a | 0.174 | 0.236 |
| F | 327.1 | 12.46 |
| N | 14384 | 10893 |

This table reports the results of moderating effect of CEO pay regulation on senior executives' overseas experience. Variables are defined in Table 1. The sample period is from 2007 to 2019. In parentheses are t-statistics.

***, **, * indicate statistical significance at the 1%, 5%, and 10% level, respectively.

## The moderating role of CEO pay regulation

In the study of the impact of executives on innovative behavior, many have focused on the factor of salary payment. We refer to Li et al. [23, 24] and study the moderating role of CEO pay regulation. To test the moderating effect of CEO pay regulation on senior executives' overseas experience, the relevant model is constructed as follows:

$$
Patent_{i,t}, Citation_{i,t} = \beta_0 + \beta_1 Oversea\_Board_{i,t} + \beta_2 Rest_{i,t} \\
+ \beta_3 Oversea\_Board_{i,t} \times Rest_{i,t} + \theta X + \omega_{i,t} + \eta_t + \epsilon_{i,t},
$$

(2)

In Eq (2), the moderating variable is $Rest_{i,t}$(calculated as natural logarithm of the pay gap between the top three executives and that of all other executives). The definitions of other variables and symbols are consistent with Eq (1).

Table 10 reports these results based on Eq (2). We focus on the coefficients of the interaction term of *Oversea_Board* and *Rest*. The results show that the coefficients of the interaction term in columns (1) and (2) are significantly negative. The CEO pay regulation have reduced the effect of senior executives' overseas experience on promoting green innovation. In addition, the coefficient of *Rest* is negative, indicating that the CEO pay regulation themselves have a negative impact on green innovation, which is similar to the research of Li et al. [24].

## Robustness test

Overall, the overseas experience of executives has indeed enhanced the green innovation of the enterprise. However, the subsample analysis showed significant heterogeneity, does this make our results unreliable? We need to conduct a series of robustness tests to prove the reliability. Specifically, we conducted an analysis from the following five aspects.

**Table 11. Robustness test: Fixed-effect model.**

|  | (1) | (2) |
|---|---|---|
|  | *Patent* | *Citation* |
| *Oversea_Board* | 0.126*** | 0.219** |
|  | (2.88) | (2.033) |
| Control variables | Yes | Yes |
| *Constant* | Yes | Yes |
| Year fixed effects | Yes | Yes |
| r2_a | 0.19 | 0.20 |
| F | 42.38 | 20.90 |
| N | 14413 | 10912 |

This table reports the results of fixed-effect model regressions analyzing the effect of returnees' experience on corporate green innovation. Variables are defined in Table 1. The sample period is from 2007 to 2019. In parentheses are t-statistics.

***, **, * indicate statistical significance at the 1%, 5%, and 10% level, respectively.

## Fixed-effect model

In addition to the main regression, this study further explored the inherent characteristics of enterprises, which might affect a company's green innovation. Generally, factors such as the unique corporate culture, market position, and government support of individual enterprises can significantly affect their innovation behavior. Therefore, we use a fixed effects model to control individual effects to eliminate this impact. Table 11 shows the regression results of the fixed-effects model. The *Oversea_Board* coefficients were significantly positive at the 1% and 5% levels, basically consistent with the sign and significance of the main regression results.

## IV-2SLS

To further address the endogeneity problem, instrumental variable was selected for regression. Following Ang et al. [25] the number of Western-style universities established by Christian missionaries in various regions of China as of 1920 was selected as an instrumental variable. The impact of education is long-lasting. In 1921, there were only 13 national and private universities in China, while the rest were established by foreign missions. This has led to a more popular trend of studying abroad in regions with missions universities, and this influence has lasted for decades. We set a dummy variable, clone, which is 1 if a province established a foreign missions university in 1921, otherwise it is 0. Specifically, we use a two-stage least squares method for regression analysis. Instrumental variables (*clone*) and executives' overseas experience (*Oversea_Board*) were applied to the regression model. Table 12 shows the results. The regression results of the first stage are shown in column (1). The coefficient of *clone* is significantly positive. The regression results of the second stage are shown in columns (2) and (3). Similarly, The coefficient of the *Oversea_Board* is significantly positive.. Thus, after eliminating potential endogeneity problems, the results remained robust.

## Tobit regression

Some companies had zero green R&D patents. If ordinary least squares (OLS) is used to perform linear regression on the overall sample, the nonlinear part will be included in the disturbance term, resulting in inconsistent regression results. To further ensure the validity of the regression results, this study used the tobit model for regression. Table 13 shows the results.

**Table 12. Robustness test: Instrumental variable method (2SLS).**

|  | (1) |  | (2) |  |
|---|---|---|---|---|
|  | *Oversea_Board* |  | *Patent* | *Citation* |
| *clone* | 0.055*** |  |  |  |
|  | (2.86) |  |  |  |
| *Oversea_Board* |  |  | 3.332*** | 1.206*** |
|  |  |  | (2.79) | (8.68) |
| Control variables | Yes |  | Yes | Yes |
| *Constant* | Yes |  | Yes | Yes |
| Year fixed effects | Yes |  | Yes | Yes |
| Industry fixed effects | Yes |  | Yes | Yes |
| r2_a | 0.47 |  | -7.24 | -0.59 |
| F | 32.08 |  | 7.827 | 32.79 |
| *N* | 14413 |  | 14413 | 10912 |
| Underidentification test (LM statistic) |  |  | 139.41 | 188.32 |
| Weak identification test (F statistic) |  |  | 70.35 | 95.14 |
| Overidentification test (Sargan statistic) |  |  | 0.39 | 0.04 |

This table reports the results of 2SLS regressions analyzing the effect of returnees' experience on corporate green innovation based on instrumental variable. Variables are defined in Table 1. The sample period is from 2007 to 2019. In parentheses are t-statistics.

***, **, * indicate statistical significance at the 1%, 5%, and 10% level, respectively.

The *Oversea_Board* coefficient was significantly positive at the 5% and 1% levels, basically consistent with the sign and significance of the main regression results.

## Replace the independent variable

In this study, *Oversea_Board* was the key independent variable of the main regression. This indicator can accurately determine whether there are people on the senior management team with overseas experience and analyze the effect of different types of overseas experience on a company's green innovation. However, this indicator cannot effectively measure the structure of the senior management team. Therefore, this study replaced the independent variable with

**Table 13. Robustness test: Tobit regression.**

|  | (1) | (2) |
|---|---|---|
|  | *Patent* | *Citation* |
| *Oversea_Board* | 0.126** | 0.167*** |
|  | (2.04) | (2.91) |
| Control variables | Yes | Yes |
| *Constant* | Yes | Yes |
| Year fixed effects | Yes | Yes |
| Industry fixed effects | Yes | Yes |
| F | 20.69 | 18.88 |
| *N* | 14413 | 10912 |

This table reports the results of Tobit regressions analyzing the effect of returnees' experience on corporate green innovation. Variables are defined in Table 1. The sample period is from 2007 to 2019. In parentheses are t-statistics.

***, **, * indicate statistical significance at the 1%, 5%, and 10% level, respectively.

**Table 14. Robustness test: Number and proportion of managers with overseas experience.**

| | (1) | (2) | (3) | (4) |
|---|---|---|---|---|
| | *Patent* | *Citation* | *Patent* | *Citation* |
| *OverseaNum* | 0.087*** | 0.071*** | | |
| | (4.33) | (2.66) | | |
| *OverseaRatio* | | | 0.660*** | 0.535** |
| | | | (4.00) | (2.42) |
| Control variables | Yes | Yes | Yes | Yes |
| *Constant* | Yes | Yes | Yes | Yes |
| Year fixed effects | Yes | Yes | Yes | Yes |
| Industry fixed effects | Yes | Yes | Yes | Yes |
| r2_a | 0.17 | 0.17 | 0.17 | 0.17 |
| F | 41.54 | 19.92 | 41.40 | 19.91 |
| *N* | 14413 | 10912 | 14413 | 10912 |

This table reports the results of OLS regressions analyzing the effect of returnees' experience on corporate green innovation based on number and ratio variables. Variables are defined in Table 1. The sample period is from 2007 to 2019. In parentheses are t-statistics.

***, **, * indicate statistical significance at the 1%, 5%, and 10% level, respectively.

the number and proportion of senior managers with overseas experience and conducted a robustness test. Table 14 shows the regression results. The independent variable in columns (1) and (2) is the number of executives with overseas experience (*OverseaNum*). The *OverseaNum* coefficient was significantly positive at the 1% level. This indicates that the larger the number of executives with overseas experience, the stronger the company's green innovation ability. The independent variable in columns (3) and (4) is the proportion of senior executives with overseas experience (*OverseaRatio*). The *OverseaRatio* coefficient was significantly positive at the 1% and 5% levels. This indicates that among senior management teams, the higher the proportion of those with overseas experience, the stronger the company's green innovation capability.

## Consider the impact of the global financial crisis

Our sample period includes the subprime crisis. To avoid the interference of the crisis on the results, we set the sample period to 2010–2019. Table 15 reports these results. The *Oversea_-Board* coefficient was significantly positive at the 5% and 1% levels, basically consistent with the sign and significance of the main regression results.

## Conclusions and policy recommendations

Few studies have specifically explored the effect of executives' overseas experience on corporate green innovation. Taking 2007–2019 data for listed A-share companies in China, we evaluated the effect of executives' overseas experience on corporate green innovation, considering differences in company ownership, region, and technological content. Green innovation was found to be significantly promoted in enterprises whose executives had overseas experience, and the effect was more significant for private enterprises, those in the eastern region, and those in high-tech industries. Another interesting fact is that CEO pay regulation have a negative moderating effect on this effect. Moreover, this positive effect is mainly reflected in senior executives. For robustness, we used a fixed effects regression model, instrumental variable two-stage

**Table 15. Robustness test: Consider the impact of the global financial crisis.**

|  | (1) | (2) |
|---|---|---|
|  | *Patent* | *Citation* |
| *Oversea_Board* | 0.0960** | 0.171*** |
|  | (2.19) | (2.92) |
| Control variables | Yes | Yes |
| *Constant* | Yes | Yes |
| Year fixed effects | Yes | Yes |
| Industry fixed effects | Yes | Yes |
| r2_a | 0.17 | 0.16 |
| F | 41.33 | 20.62 |
| *N* | 13457 | 10457 |

This table reports the results of ols regressions analyzing the effect of returnees' experience on corporate green innovation. Variables are defined in Table 1. The sample period is from 2010 to 2019. In parentheses are t-statistics. ***, **, * indicate statistical significance at the 1%, 5%, and 10% level, respectively.

least squares (IV-2SLS) regression, and Tobit regression, and replaced the 0–1 independent variables with number and ratio variables.

We supplement the literature on the effects of executives' overseas experience while also further enriching upper echelons theory. We also expand theoretical research related to green finance; it especially provides a theoretical basis for emerging market economies to promote green innovation by introducing overseas talent. In addition, this research has strong practical significance. It provides data and evidence to help governments propose industrial and regional policies for emerging enterprises. Policies focused on introducing overseas talent in China's western region and improving the incentive system of SOEs can support making full use of the promotion effect of overseas talent on green innovation. Of course, pay (or salary) regulation are also crucial in corporate governance.

## Supporting information

**S1 Data.**
(XLSX)

## Author Contributions

**Conceptualization:** Fangze Cheng.

**Data curation:** Fangze Cheng, Xin Kuang.

**Formal analysis:** Fangze Cheng, Xin Kuang.

**Validation:** Xin Kuang.

**Writing – original draft:** Fangze Cheng.

**Writing – review & editing:** Xin Kuang.

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
