## [Decision Letter · Decision Letter 0]

18 Aug 2022

PONE-D-22-21551Effect of Senior Executives’ Overseas Experience on Corporate Green InnovationPLOS ONE

Dear Dr. fangze,

Thank you for submitting your manuscript to PLOS ONE. After careful consideration, we feel that it has merit but does not fully meet PLOS ONE’s publication criteria as it currently stands. Therefore, we invite you to submit a revised version of the manuscript that addresses the points raised during the review process.

ACADEMIC EDITOR:The reviewers are experts in the area and have prepared a careful and fair review. I personally appreciate your efforts in writing the manuscript and I find the topic very interesting and worth pursuing. As you can see from the report, the reviewers have provided some specific, constructive and clear comments for you to improve the quality of the paper. I have also considered it myself taking into account the comments from reviewers and decided to give you an opportunity to revise your manuscript. I hope you will take this chance to improve the quality of the paper to satisfy both reviewers. Please carefully follow the comments to revise your manuscript and resubmit it for re-consideration for publication. You can find detailed comments in the review reports. As those comments are presented in a clear way, so I will not repeat them here to avoid your confusions. 

We look forward to receiving your revised manuscript.

Kind regards,

Vu Quang Trinh, PhD

Academic Editor

PLOS ONE

Journal Requirements:

Additional Editor Comments:

NA

Reviewers' comments:

Reviewer's Responses to Questions

**Comments to the Author**

1. Is the manuscript technically sound, and do the data support the conclusions?

Reviewer #1: Partly

Reviewer #2: Yes

2. Has the statistical analysis been performed appropriately and rigorously? 

Reviewer #1: N/A

Reviewer #2: Yes

3. Have the authors made all data underlying the findings in their manuscript fully available?

Reviewer #1: No

Reviewer #2: Yes

4. Is the manuscript presented in an intelligible fashion and written in standard English?

Reviewer #1: Yes

Reviewer #2: Yes

5. Review Comments to the Author

Reviewer #1: Referee report

Manuscript ID PONE-D-22-21551

Effect of Senior Executives’ Overseas Experience on Corporate Green Innovation

This current study investigates how senior executives’ overseas experience (studying, living, and working) on corporate green innovation in China over the period from 2007 to 2019. The author suggests that senior executives’ overseas experience can promote corporate green innovation measured by the number of patents and citations. Further analyses indicate the variations of the effect on the cross-sections.

Overall, the paper offers several interesting findings and implications for practitioners. However, the introduction and literature review are immature and need serious care. My comments and suggestions are as follows:

1. The introduction: I do not think this is a proper Introduction part. With just two paragraphs, the introduction fails to address the gap in the literature and does not show why the study matters. I strongly recommend the author to follow this guide to prepare the introduction:

http://marcfbellemare.com/wordpress/wp-content/uploads/2020/09/BellemareHowToPaperSeptember2020.pdf

2. The hypotheses development: Currently immature and unconvincing. H2, H3, and H4 lack a clear rationale and discussion of literature for why there are differences in the effect across different categories of firms. Any supporting evidence? Any theory behind the hypotheses? The author should elaborate more on his/her reasoning for the hypotheses.

3. How does the effect change with time? Specifically, do CEOs who came back to China 20 years ago differ from whom came back just recently?

4. Subsample regressions: please provide the coefficient difference test of Oversea_Board’s coefficients across different subsample regressions in Tables 6, 7, and 8. If the coefficients are not statistically significant, then the findings of H2, H3, and H4 analysis are not valid.

5. Why are the R-squares negative in Columns 2 and 3 in Table 11? Generally, a negative R-sq means the model does not fit well with the data. Therefore, there might be potential misspecification in the model setting. Please consider this issue seriously.

6. IV regressions: Please report the results of the diagnostic tests of IV regression: under-identification test, weak-identification test, and over-identification test. If the IV does not meet the test requirements, then IV regression results are not valid.

Because I find the empirical findings interesting, I would recommend a risky major revision. Please carefully follow the comments and address them one by one. Thank you and I look forward to receiving your revised manuscript.

Reviewer #2: Summary

The paper studies how senior executives’ oversea experience influences corporate green innovation. By collecting yearly data of listed A-share companies in China from 2007 to 2019, the data sample contains 15,389 panel data of 2,681 companies. The results showed that senior executives’ overseas experience could promote green innovation in companies. This positive effect was more significant for private enterprises and high-tech enterprises, especially in Eastern China. This study enriches upper echelons theory and provides theoretical support for government agencies to accelerate innovative green development strategies

Please read the attached document.

6. PLOS authors have the option to publish the peer review history of their article (what does this mean?). If published, this will include your full peer review and any attached files.

Reviewer #1: No

Reviewer #2: No

---

## [Author Response · Author response to Decision Letter 0]

7 Oct 2023

Response to reviewer comments

Dear Reviewer,

We are very grateful for your favorable view on our manuscript. Your comments are very important to the acceptability of our paper. According to your suggestions, we have made the following revisions on this manuscript: 

General comments:

●The topic is nice and worthwhile studying. The contribution that this paper aims to present is also noticeable when examining the impact of oversea experience of senior executives on the corporate green innovation.

●The time chosen for the data sample is from 2007 to 2019. The author(s) might have intendedly neglected the potential bias from the global financial crisis (2008/2009). Thus, I would consider that the author(s) should consider the impact of the global financial crisis on data analysis.

Response：

Indeed, there is a lack of summary of our contributions to the paper. Based on your suggestion, we have added a description of our contributions at the end of the introduction.

In order to avoid potential biases caused by the global financial crisis, we controlled for the impact of this factor in the robustness testing section. The specific content is as follows:

Major comments:

●The research background and research motivation are not clear. For example, many studies (???) have investigated the effect of executives’ personal characteristics and background on corporate innovation. 

Response：

Our background and motivation are mainly based on green and innovation, and currently the academic community mostly studies the two separately, without analyzing them in a unified structure. We have added an explanation to this in the second paragraph of the introduction.

●For literature review, the author(s) have not introduced background of corporate green innovation and theories related to the relationship between executive’s overseas experience and corporate green innovation. Even though the study mentioned upper echelons theory, the author(s) can think of applying more theories like imprinting theory following previous studies (He, Chen & Zhang, 2021; Quan, Quian & Zhang, 2021). 

Response：

We have added existing research on imprinting theory in the literature review section.

●I think that the paper may need check data sample. The author(s) used a sample of 15,389 observations from 2,681 companies. However, according to the descriptive statistics, the highest observations of variables is less than 15,389. The author(s) should double check the data sample.

Response：

We have checked the data sample and corrected the observations. Now, our sample observations is 15334. It should be noted that some variables have missing observations, so not all variables have observations of 15334.

●I am just wondering how the author(s) define executives? Do senior executives include chairman, CEO, COO…? The author(s) divided executives to sub-sample including senior managers and junior managers. I suppose that chairman and/or CEO’s oversea experience have strong impact on corporate green innovation. The author(s) can think of investigating the moderating role of chairman and CEO. Furthermore, the author(s) should think of the title: senior executives or executives or senior managers or managers.

Response：

We define executives according to China's Company Law, which includes the chairman, CEO, and COO. Indeed, our empirical evidence also indicates that the impact of senior managers is significant and more important as you mentioned.

Examining the moderating effect of senior managers is indeed a good suggestion. Unfortunately, multicollinearity may be encountered when conducting regulatory effect analysis. The overseas experiences of executives themselves are divided into two categories: senior executives and junior executives, and the interaction term may be omitted in regression process, so we did not conduct this analysis. However, don't worry, our results have shown that senior executives are more influenced.

●The author(s) used panel data. However, the model did not show the panel data. Furthermore, fixed effect model (FEM) is a statistic are technique to analyse data. The author(s) should explain why the author(s) use FEM to analyse data rather than adding fixed effects to the research model.

Response：

We have edited the model again, as shown in Eq (1). 

In the robustness testing section, we also added explanations on why FEM is used.

●The author(s) described the results well. However, the discussion is not clearly. The author(s) should focus on how the findings link with theories and previous studies.

Response：

We have added a connection with previous theories and studies in the explanation of the results.

●The author(s) did many robust tests. However, the author(s) should use sub-sample to explain the reasons for using these robust tests. 

Response：

We have used sub-sample to explain the reasons for using these robust tests. The specific content is as follows:

Overall, the overseas experience of executives has indeed enhanced the green innovation of the enterprise. However, the sub-sample analysis showed significant heterogeneity, does this make our results unreliable? We need to conduct a series of robustness tests to prove the reliability. 

Minor points:

●The author(s) may need to introduce correlation matrix.

Response：

We have added a correlation matrix.

●The author(s) can consider GMM/2SLS for endogeneity. 

Response：

We have selected instrumental variables in the robustness testing section and conducted 2SLS analysis. This greatly solves endogeneity.

---

## [Decision Letter · Decision Letter 1]

17 Nov 2023

PONE-D-22-21551R1Effect of Senior Executives’ Overseas Experience on Corporate Green InnovationPLOS ONE

Dear Dr. Kuang,

Thank you for submitting your manuscript to PLOS ONE. After careful consideration, we feel that it has merit but does not fully meet PLOS ONE’s publication criteria as it currently stands. Therefore, we invite you to submit a revised version of the manuscript that addresses the points raised during the review process.

We look forward to receiving your revised manuscript.

Kind regards,

Rana Muhammad Ammar Zahid, PhD

Academic Editor

PLOS ONE

Journal Requirements:

Reviewers' comments:

Reviewer's Responses to Questions

**Comments to the Author**

1. If the authors have adequately addressed your comments raised in a previous round of review and you feel that this manuscript is now acceptable for publication, you may indicate that here to bypass the “Comments to the Author” section, enter your conflict of interest statement in the “Confidential to Editor” section, and submit your "Accept" recommendation.

Reviewer #1: (No Response)

Reviewer #3: (No Response)

2. Is the manuscript technically sound, and do the data support the conclusions?

Reviewer #1: Yes

Reviewer #3: Yes

3. Has the statistical analysis been performed appropriately and rigorously? 

Reviewer #1: Yes

Reviewer #3: Yes

4. Have the authors made all data underlying the findings in their manuscript fully available?

Reviewer #1: Yes

Reviewer #3: Yes

5. Is the manuscript presented in an intelligible fashion and written in standard English?

Reviewer #1: Yes

Reviewer #3: Yes

6. Review Comments to the Author

Reviewer #1: Overall, the author(s) addressed my comments well. I see a great improvement in the manuscript following this revision. I would like to thank the author(s) for taking my comments and suggestions seriously.

At the current state of the manuscripts, I see that some more clarification and editing are needed for the paper to meet the publication standards of the journal. Please find my points listed as follows:

1. Introduction section is still immature. You need to clearly layout the introduction section in a way that it clearly summarizes what you did in this study, what is the research gap, what you found, then the contribution, and the brief structure of the paper. Please follow the following great paper by Professor Bellemare to restructure and develop your Introduction section more. I am sure it will benefit you.

http://marcfbellemare.com/wordpress/wp-content/uploads/2020/09/BellemareHowToPaperSeptember2020.pdf

2. Explanation of the reason why choosing the IV: How the chosen IV meet the exclusion restrictions of IV/2SLS estimation? If not explaining it carefully, the IV is not valid. Moreover, the author(s) must provide the statistics of the under-identification and weak identification tests for the IV-2SLS estimation. It is mandatory when use IV/2SLS regression.

3. Numbering sections: The current section and sub-section structure is hard to follow. Please number them consecutively.

4. Presentation of the paper: I see numerous grammar and typos in the manuscript. For examples:

- Data section: Data collected from "Cathay Pacific"? I think you meant CSMAR?

- "Table10", "Table11" with no space between "Table" and the table number.

Reviewer #3: Comment to author

The author/s/ of the paper seem to have made significant improvements based on the feedback received during the first round of revisions. However, upon conducting a meticulous examination of the paper, it becomes apparent that the study primarily focuses on the relationship between overseas executives and green innovation (GI), a topic that has already been extensively discussed and debated in the existing literature. This review points out the need for further refinement and expansion of the paper's content, particularly in the area of exploring the factors that drive green innovation beyond the involvement of foreign executives.

One notable observation is that the paper concludes that green innovation can be driven by the presence of foreign executives (GI could be driven by having foreign executive GI). While this conclusion has merit, it's crucial to consider the broader context and the numerous variables that influence green innovation. There's a rich body of literature that links green innovation to CEO and executive actions, specifically related to CEO pay regulation. Surprisingly, the author(s) have omitted this critical aspect from their hypotheses and subsequent analysis.

To address this omission effectively, the author(s) should include a subsection dedicated to the drivers of green innovation within the paper. This subsection should delve into the research on the influence of CEOs and executives, as well as the role of CEO pay regulation in promoting or hindering green innovation initiatives. By incorporating this aspect, the paper would be more comprehensive and aligned with the current state of research in this field. To further enhance the study's comprehensiveness, the author should consider including the following relevant studies:

• Li, Q., Maqsood, U. S., & Zahid, R. M. A. (2023). "Nexus between government surveillance on executive compensation and green innovation: Evidence from the type of state-owned enterprises." Published in Business Ethics, the Environment & Responsibility, this study provides insights into the relationship between government oversight of executive compensation and green innovation.

• Li, Q., Maqsood, U.S., Zahid, R.A., et al. "Regulating CEO pay and green innovation: moderating role of social capital and government subsidy" (2023). Published in the Environmental Science and Pollution Research, this study explores the regulatory aspects of CEO pay and its impact on green innovation.

These recent studies are directly related to the topic at hand and have formulated their hypotheses accordingly.

One major concern raised by the reviewer in R1 is regarding the IV-2SLS. The author claims to have used the term "clone," but it is unclear how this term is defined and how it relates to the current scenario in the context of green innovation? I strongly recommend that the author addresses these points seriously and revises the manuscript with a more comprehensive exploration of incentives and the suggested improvements to strengthen the narrative.

7. PLOS authors have the option to publish the peer review history of their article (what does this mean?). If published, this will include your full peer review and any attached files.

Reviewer #1: No

Reviewer #3: **Yes: **Umer Sahil Maqsood

---

## [Author Response · Author response to Decision Letter 1]

14 Dec 2023

Response to reviewer #1 comments

Dear Reviewer #1,

We are very grateful for your favorable view on our manuscript. Your comments are very important to the acceptability of our paper. According to your suggestions, we have made the following revisions on this manuscript: 

1. Introduction section is still immature. You need to clearly layout the introduction section in a way that it clearly summarizes what you did in this study, what is the research gap, what you found, then the contribution, and the brief structure of the paper. Please follow the following great paper by Professor Bellemare to restructure and develop your Introduction section more. I am sure it will benefit you.

http://marcfbellemare.com/wordpress/wp-content/uploads/2020/09/BellemareHowToPaperSeptember2020.pdf

Response 1：

We have carefully read Professor Bellemare's paper, and based on your suggestions and Professor Bellemare's views, we have edited the introduction section again.

2. Explanation of the reason why choosing the IV: How the chosen IV meet the exclusion restrictions of IV/2SLS estimation? If not explaining it carefully, the IV is not valid. Moreover, the author(s) must provide the statistics of the under-identification and weak identification tests for the IV-2SLS estimation. It is mandatory when use IV/2SLS regression.

Response 2：

We have added relevant explanations in the IV-2SLS section and added the statistics of the under-identification and weak identification tests.

3. Numbering sections: The current section and sub-section structure is hard to follow. Please number them consecutively.

Response 3：

We have reconfirmed the coherence between section and subsection.

4. Presentation of the paper: I see numerous grammar and typos in the manuscript. For examples:

- Data section: Data collected from "Cathay Pacific"? I think you meant CSMAR?

- "Table10", "Table11" with no space between "Table" and the table number.

Response 4：

We carefully checked and corrected grammar and typos.

Response to reviewer #3 comments

Dear Reviewer #3,

We really appreciate you for your carefulness and conscientiousness. Your suggestions are really valuable and helpful for revising and improving our paper. According to your suggestions, we have made the following revisions on this manuscript: 

1. The author/s/ of the paper seem to have made significant improvements based on the feedback received during the first round of revisions. However, upon conducting a meticulous examination of the paper, it becomes apparent that the study primarily focuses on the relationship between overseas executives and green innovation (GI), a topic that has already been extensively discussed and debated in the existing literature. This review points out the need for further refinement and expansion of the paper's content, particularly in the area of exploring the factors that drive green innovation beyond the involvement of foreign executives.

One notable observation is that the paper concludes that green innovation can be driven by the presence of foreign executives (GI could be driven by having foreign executive GI). While this conclusion has merit, it's crucial to consider the broader context and the numerous variables that influence green innovation. There's a rich body of literature that links green innovation to CEO and executive actions, specifically related to CEO pay regulation. Surprisingly, the author(s) have omitted this critical aspect from their hypotheses and subsequent analysis.

To address this omission effectively, the author(s) should include a subsection dedicated to the drivers of green innovation within the paper. This subsection should delve into the research on the influence of CEOs and executives, as well as the role of CEO pay regulation in promoting or hindering green innovation initiatives. By incorporating this aspect, the paper would be more comprehensive and aligned with the current state of research in this field. To further enhance the study's comprehensiveness, the author should consider including the following relevant studies:

• Li, Q., Maqsood, U. S., & Zahid, R. M. A. (2023). "Nexus between government surveillance on executive compensation and green innovation: Evidence from the type of state-owned enterprises." Published in Business Ethics, the Environment & Responsibility, this study provides insights into the relationship between government oversight of executive compensation and green innovation.

• Li, Q., Maqsood, U.S., Zahid, R.A., et al. "Regulating CEO pay and green innovation: moderating role of social capital and government subsidy" (2023). Published in the Environmental Science and Pollution Research, this study explores the regulatory aspects of CEO pay and its impact on green innovation.

These recent studies are directly related to the topic at hand and have formulated their hypotheses accordingly.

Response 1：

We have added a section that references the research of Li et al. (2023a,2023b), specifically designed to examine the moderating effect of CEO pay regulation. The specific content is as follows:

2. One major concern raised by the reviewer in R1 is regarding the IV-2SLS. The author claims to have used the term "clone," but it is unclear how this term is defined and how it relates to the current scenario in the context of green innovation? I strongly recommend that the author addresses these points seriously and revises the manuscript with a more comprehensive exploration of incentives and the suggested improvements to strengthen the narrative.

Response 2：

We have added relevant explanations in the IV-2SLS section. Furthermore, we have strengthened the narrative of the manuscript.

References:

Li Q, Maqsood US, Zahid RMA. Nexus between government surveillance on executive compensation and green innovation: Evidence from the type of state-owned enterprises.Business Ethics, the Environment & Responsibility. 2023a. https://doi.org/10.1111/beer.12601

Li Q, Maqsood US, Zahid RMA, Anwar W.Regulating CEO pay and green innovation: moderating role of social capital and government subsidy. Environmental Science and Pollution Research. 2023b. https://doi.org/10.1007/s11356-023-26641-x

---

## [Editor Report · Decision Letter 2]

18 Dec 2023

Effect of Senior Executives’ Overseas Experience on Corporate Green Innovation

PONE-D-22-21551R2

Dear Dr. Kuang,

We’re pleased to inform you that your manuscript has been judged scientifically suitable for publication and will be formally accepted for publication once it meets all outstanding technical requirements.

Kind regards,

Rana Muhammad Ammar Zahid, PhD

Academic Editor

PLOS ONE

Additional Editor Comments (optional):

Thank you for incorporating suggested changes
---

## [Editor Report · Acceptance letter]

24 Apr 2024

PONE-D-22-21551R2 

PLOS ONE

Dear Dr. Kuang, 

I'm pleased to inform you that your manuscript has been deemed suitable for publication in PLOS ONE. Congratulations! Your manuscript is now being handed over to our production team.

Kind regards, 

on behalf of

Dr. Rana Muhammad Ammar Zahid 

Academic Editor

PLOS ONE